# Noise Map Guidance: Inversion with Spatial Context for Real Image Editing

**Hansam Cho**[1,2*], **Jonghyun Lee**[1,2], **Seoung Bum Kim**[1], **Tae-Hyun Oh**[3,4†], **Yonghyun Jeong**[2†]

[1]School of Industrial and Management Engineering, Korea University, [2]NAVER Cloud
[3]Dept. of Electrical Engineering and Grad. School of Artificial Intelligence, POSTECH
[4]Institute for Convergence Research and Education in Advanced Technology,Yonsei University
{chosam95, tomtom1103, sbkim1}@korea.ac.kr
{taehyun}@postech.ac.kr,{yonghyun.jeong}@navercorp.com

## ABSTRACT

Text-guided diffusion models have become a popular tool in image synthesis, known for producing high-quality and diverse images. However, their application to editing *real* images often encounters hurdles primarily due to the text condition deteriorating the reconstruction quality and subsequently affecting editing fidelity. Null-text Inversion (NTI) has made strides in this area, but it fails to capture spatial context and requires computationally intensive per-timestep optimization. Addressing these challenges, we present NOISE MAP GUIDANCE (NMG), an inversion method rich in a spatial context, tailored for real-image editing. Significantly, NMG achieves this without necessitating optimization, yet preserves the editing quality. Our empirical investigations highlight NMG's adaptability across various editing techniques and its robustness to variants of DDIM inversions.

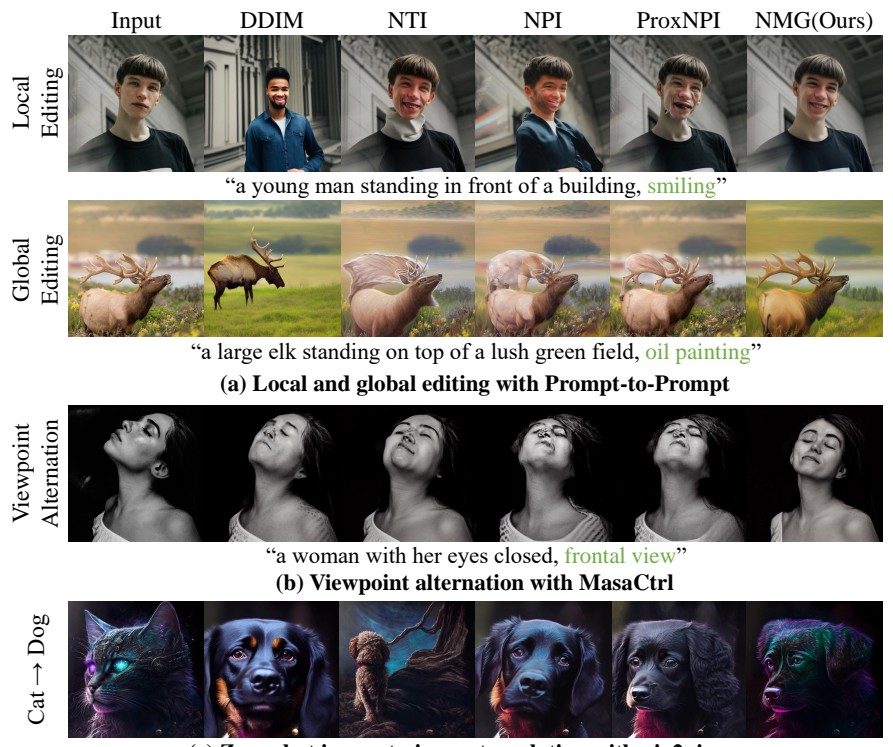

(a) **Local and global editing with Prompt-to-Prompt**

(b) **Viewpoint alternation with MasaCtrl**

(c) **Zero-shot image-to-image translation with pix2pix-zero**

Figure 1: Compared to other inversion methods, NMG (a) demonstrates high fidelity editing when paired with Prompt-to-Prompt, (b) successfully conducts viewpoint alternation via MasaCtrl, and (c) preserves the spatial context of the input image while performing zero-shot image-to-image translation with pix2pix-zero. Text prompt corresponding to each input image is presented beneath each sample, with words introduced for image editing distinctly highlighted in green.

---

[*]First Author. Work done during an internship at NAVER Cloud.
[†]Co-corresponding authors.

# 1 INTRODUCTION

Text-guided diffusion models (Rombach et al., 2022; Saharia et al., 2022; Ramesh et al., 2022; Chang et al., 2023; Podell et al., 2023) have recently emerged as a powerful tool in image synthesis, widely adapted for their ability to generate images of exceptional visual quality and diversity. Owing to their impressive performance, numerous studies have leveraged these models for image editing (Hertz et al., 2022; Cao et al., 2023; Parmar et al., 2023; Epstein et al., 2023). While these models excel in editing synthesized images, they often produce sub-par results when editing real images. Hertz et al. (2022) point out that this challenge mostly originates from the reliance on classifier-free guidance (CFG) (Ho & Salimans, 2021), a method ubiquitously adopted to increase fidelity

Editing real images within a diffusion framework follows a twofold approach. Firstly, an image is inverted into a latent variable via DDIM inversion. This latent variable is then channeled into two paths: reconstruction and editing. While the editing path modifies the latent towards the desired outcome, it also integrates information from the reconstruction path to retain the original identity of the image. As a result, the quality of the edit is highly dependent on the reconstruction path. However, extrapolating towards the condition via CFG introduces errors in each timestep. This deviation pushes the reconstruction path away from the DDIM inversion trajectory, hindering accurate reconstruction and ultimately diminishing the editing fidelity.

Addressing this limitation, Mokady et al. (2023) propose Null-text Inversion (NTI) leading to notable achievements in real image editing. This approach optimizes the null-text embedding used in CFG on a per-timestep basis, correcting the reconstruction path to the desired DDIM inversion trajectory. By leveraging the optimized null-text embedding and integrating it with Prompt-to-Prompt (Hertz et al., 2022) editing, NTI conducts real image editing. While NTI offers certain benefits, its use of per-timestep optimization can lead to increased computational demands in practical applications. As a solution to this time-intensive optimization method, NPI (Miyake et al., 2023) and ProxNPI (Han et al., 2023) attempt to approximate the optimized null-text embedding without optimization. Although these methods succeed in correcting the reconstruction path, they often struggle to capture the spatial context in the input images.

To address these challenges, we introduce NOISE MAP GUIDANCE (NMG), an inversion methodology enriched with spatial context for real image editing. To capture spatial context, we employ the latent variables from DDIM inversion, which we refer to as *noise maps*. These noise maps, essentially noisy representations of images, inherently encapsulate the spatial context. To eliminate the need for an optimization phase, we condition the noise maps to the reverse process. Rather than solely depending on text embeddings for image editing, our methodology harnesses both noise maps and text embeddings, drawing from their spatial and semantic guidance to perform faithful editing. To accommodate our dual-conditioning strategy, we reformulate the reverse process by leveraging the guidance technique proposed by Zhao et al. (2022).

Our experimental results highlight NMG's capacity to preserve the spatial context of the input image during real image editing. Figure 1 (a) and (c) reveal that NTI, NPI, and ProxNPI often struggle to capture the spatial context of the input image. Furthermore, Figure 1 (b) highlights a scenario where spatial context is essential for effective editing, and in such context, NMG consistently outperforms other methods. By utilizing guidance techniques of diffusion models, our optimization-free method achieves speeds substantially faster than NTI without compromising editing quality. The versatility of NMG is further emphasized by its integration with various editing techniques, e.g., Prompt-to-Prompt (Hertz et al., 2022), MasaCtrl (Cao et al., 2023) and pix2pix-zero (Parmar et al., 2023), each grounded in an inversion methodology. Moreover, we demonstrate NMG's resilience to variations of DDIM inversion. Our main contributions are summarized as follows:

- We present NOISE MAP GUIDANCE (NMG), an inversion method rich in spatial context, specifically tailored for real-image editing.
- Although formulated as an optimization-free method, we show that NMG maintains editing quality without compromise, even achieving superior performance over competing methods.
- We demonstrate NMG's adaptability by combining it with different editing methodologies and by highlighting its consistent robustness across variations of DDIM inversion.

## 2 RELATED WORK

**Inversion with Diffusion Models**   The initial stage of image editing typically consists of encoding the input image into a latent space, referred to as inversion. DDIM (Song et al., 2020a) first introduces the concept of inversion within diffusion models by formulating the sampling process as an ordinary differential equation (ODE) and inverting it to recover a latent noise as a starting point for the sampling process. However, the precision of DDIM's reconstruction deteriorates when being integrated with a text-guided diffusion model (Hertz et al., 2022). In response, recent studies propose the addition of an auxiliary diffusion state (Wallace et al., 2023) or a different inversion framework (Huberman-Spiegelglas et al., 2023). Recently, Mokady et al. (2023) proposes Null-text Inversion (NTI), which tailors inversion techniques specifically for text-guided diffusion models. NTI refines the null-text embedding used in text-guided diffusion models, achieving accurate image reconstruction. However, the intensive computation required for optimization constrains its broader applicability. To mitigate this drawback, Negative-prompt Inversion (NPI) (Miyake et al., 2023) approximates the optimized null-text embedding to achieve optimization-free inversion, albeit with compromised performance relative to NTI. Subsequently, Han et al. (2023) incorporate proximal guidance into NPI, enhancing its quality. Like NPI, we introduce an optimization-free inversion technique, but uniquely, our approach robustly maintains the spatial context of the original image.

**Editing with Inversion methods**   Despite the emergence of numerous studies in the field of editing via diffusion models, they often encounter challenges including the need for extra training (Kawar et al., 2023; Valevski et al., 2023; Bar-Tal et al., 2022), or the incorporation of additional conditional signals (Avrahami et al., 2022; 2023; Nichol et al., 2022; Rombach et al., 2022). Recently, inversion-based editing methods (Hertz et al., 2022; Cao et al., 2023; Parmar et al., 2023) have shown promising results that do not require additional training or conditional signals. These methods typically involve two parallel phases: a reconstruction sequence and an editing sequence. In the reconstruction sequence, critical information from the input image is extracted and fed into the editing sequence for manipulation. Notably, Prompt-to-Prompt (Hertz et al., 2022) leverages cross-attention maps to guide the editing sequence, while MasaCtrl (Cao et al., 2023) introduces mutual self-attention to facilitate non-rigid editing. pix2pix-zero (Parmar et al., 2023) utilizes cross-attention map guidance to perform zero-shot image-to-image translation. Because obtaining precise information from the reconstruction path is crucial for reliable image editing, the efficiency of these methods heavily relies on the performance of the inversion methods they use. By integrating our inversion method with existing inversion-based editing methods, we demonstrate significant improvement in preserving the spatial context of input images.

## 3 METHOD

### 3.1 BACKGROUND

**Text-guided Diffusion Model**   Text-guided diffusion models (Rombach et al., 2022) are designed to map a random Gaussian noise vector $z_T$ into an image $z_0$ while aligning with the given text condition $c_T$, typically text embeddings derived from text encoders like CLIP (Radford et al., 2021). This is achieved through a sequential denoising operation, commonly termed as the reverse process. This process is driven by a noise prediction network $\epsilon_\theta$, which is optimized by the loss:

$$L_{simple} = E_{z_0, \epsilon \sim N(0,I), t \sim U(1,T)} \| \epsilon - \epsilon_\theta(z_t, t, c_T) \|_2^2. \tag{1}$$

Although $\epsilon_\theta$ is conditioned on the timestep $t$, denoted as $\epsilon_\theta(z_t, t, c_T)$, we omit the timestep condition for the output of the network as $\epsilon_\theta(z_t, c_T)$ for brevity. Text-guided diffusion models typically utilize classifier-free guidance (CFG) (Ho & Salimans, 2021) to incorporate text conditions during image generation. CFG is represented as follows:

$$\tilde{\epsilon}_\theta(z_t, c_T) = \epsilon_\theta(z_t, \emptyset) + w \cdot (\epsilon_\theta(z_t, c_T) - \epsilon_\theta(z_t, \emptyset)). \tag{2}$$

Here, $w$ signifies the text guidance scale (controlling the impact of the text condition), and $\emptyset$ denotes the null text embedding (the embedding vector for a null-text "").

**DDIM Inversion**   To edit a real image in the framework of diffusion models, an image firstly has to be converted into a latent variable $z_T^*$, a process known as inversion. Fundamentally, the latent variable $z_T^*$ is the original starting noise that reconstructs into the image when denoised. To invert an

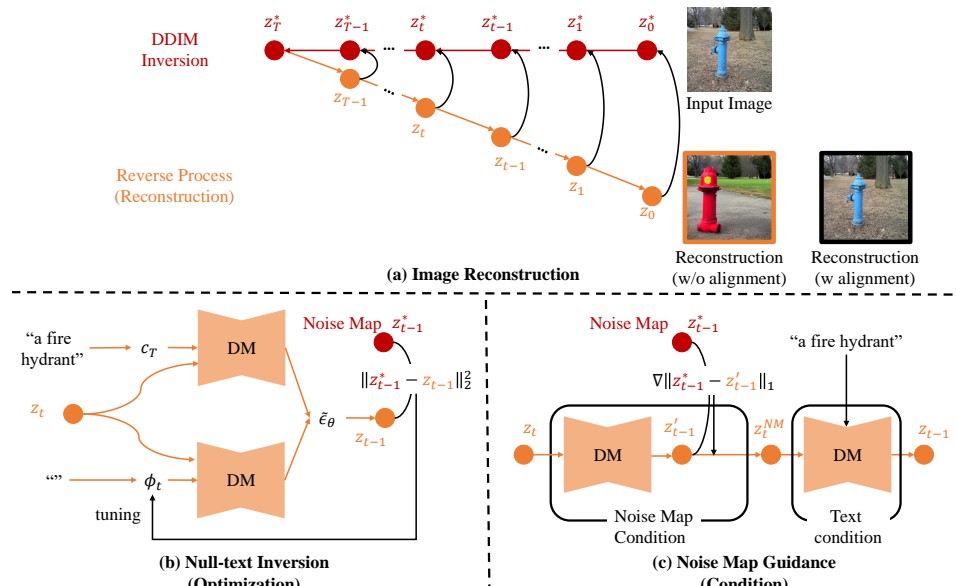

Figure 2: As seen in (a), naive reconstruction often fails due to the reconstruction path diverging from the original inversion path. Achieving reliable reconstruction necessitates realigning the reconstruction path with the inversion path. As depicted in (b), NTI achieves this alignment by optimizing the null-text embedding, thereby reducing the error between the inversion and reconstruction paths. Conversely, NMG, as shown in (c), conditions the reconstruction process based on the divergence between the two paths, leveraging this variance to refine the reconstruction path.

image into its latent variable, DDIM inversion (Song et al., 2020a) is predominantly utilized. DDIM inversion is derived from the reverse process of DDIM that is formulated as:

$$z_{t-1} = \sqrt{\frac{\alpha_{t-1}}{\alpha_t}} z_t + \sqrt{\alpha_{t-1}} \left( \sqrt{\frac{1}{\alpha_{t-1}} - 1} - \sqrt{\frac{1}{\alpha_t} - 1} \right) \epsilon_\theta(z_t, c_T) \quad (3)$$

with $\{\alpha_t\}_{t=0}^T$ as a predefined noise schedule. Because this DDIM reverse process can be formulated as an ordinary differential equation (ODE), the DDIM inversion process can be obtained by reversing said ODE as:

$$z_{t+1} = \sqrt{\frac{\alpha_{t+1}}{\alpha_t}} z_t + \sqrt{\alpha_{t+1}} \left( \sqrt{\frac{1}{\alpha_{t+1}} - 1} - \sqrt{\frac{1}{\alpha_t} - 1} \right) \epsilon_\theta(z_t, c_T). \quad (4)$$

**Null-text Inversion** Inversion-based editing methods follow a twofold approach. First, an image is converted into its latent variable through inversion. This is followed by a simultaneous two-phase procedure: reconstruction and editing. While the editing phase alters the latent variable towards the desired modification, it concurrently draws on information from the reconstruction phase to maintain the image's characteristics. During editing, a substantial CFG guidance value of $w > 1$ is crucial for generating high-fidelity images. However, the extrapolation toward the condition introduces errors at every timestep. As illustrated in Figure 2 (a), a large CFG guidance scale misguides the reconstruction path away from the inversion path. This divergence leads to an imprecise reconstructed image, subsequently degrading the final quality of the edited image. To address this, Null-text Inversion (NTI) (Mokady et al., 2023) optimizes null-text embeddings used in Eq. 2 to align with the DDIM inversion trajectory. Initially, DDIM inversion is performed with $w = 1$ to compute the latent variables $\{z_t^*\}_{t=1}^T$, termed the *noise maps*. As the reverse process progresses from $t$ to $t-1$, $z_{t-1}$ is derived from $z_t$ with Eq. 2 and Eq. 3:

$$z_{t-1} = \sqrt{\frac{\alpha_{t-1}}{\alpha_t}} z_t + \sqrt{\alpha_{t-1}} \left( \sqrt{\frac{1}{\alpha_{t-1}} - 1} - \sqrt{\frac{1}{\alpha_t} - 1} \right) \tilde{\epsilon}_\theta(z_t, c_T). \quad (5)$$

With both $z_{t-1}$ and a single noise map $z_{t-1}^*$ available, the null-text embedding $\emptyset_t$ is optimized every step using the loss function:

$$\min_{\emptyset_t} \| z_{t-1} - z_{t-1}^* \|_2^2. \quad (6)$$

By optimizing the null-text embeddings, NTI reduces the error of the reconstruction path. Overall process of NTI is depicted in Figure 2 (b). However, since the null-text embedding is represented as a single-dimensional vector in $\mathbb{R}^d$, it struggles to capture the image's spatial context.

### 3.2 NOISE MAP GUIDANCE

Unlike null-text embeddings, noise maps $\{z_t^*\}_{t=1}^T$ naturally capture the spatial context of the input image. This is due to the fact that noise maps originate from infusing a small amount of noise into the input image and have the same spatial dimensions as the input image. Leveraging this attractive trait, we directly employ noise maps to preserve the spatial context of an image. As a part of this approach, we condition the reverse process on noise maps. Given that current text-guided diffusion models are conditioned solely on text embeddings, our initial step is to reformulate the reverse process to account for both text and noise maps.

To reformulate the reverse process to a conditional form, we introduce the score function $\nabla_{z_t} \log p(z_t)$ with the relation $\epsilon_\theta(z_t) \approx -\sqrt{1-\alpha_t}\nabla_{z_t} \log p(z_t)$. By replacing $\nabla_{z_t} \log p(z_t)$ with $\nabla_{z_t} \log p(z_t|c)$, we enable the network to produce outputs based on specific conditions, leading to the equation $\epsilon_\theta(z_t, c) \approx -\sqrt{1-\alpha_t}\nabla_{z_t} \log p(z_t|c)$. Applying Bayes's rule, $\nabla_{z_t} \log p(z_t|c) = \nabla_{z_t} \log p(z_t) + \nabla_{z_t} \log p(c|z_t)$, the network's conditional output is then formulated as:

$$\epsilon_\theta(z_t, c) = -\sqrt{1-\alpha_t}(\nabla_{z_t} \log p(z_t) + \nabla_{z_t} \log p(c|z_t)) \tag{7}$$

In this context, $\nabla_{z_t} \log p(c|z_t)$ is a crucial component to condition the diffusion model. We introduce energy guidance (Zhao et al., 2022), which serves as a flexible conditioning format in our formulation. With energy guidance, Eq. 7 is revised as follows:

$$\epsilon_\theta(z_t, c) = -\sqrt{1-\alpha_t}(\nabla_{z_t} \log p(z_t) - \nabla_{z_t}\mathcal{E}(z_t, c, t)). \tag{8}$$

NTI (Mokady et al., 2023) performs one-step denoising, transitioning from $z_t$ to $z_{t-1}$, and compares the noise map $z_{t-1}^*$ with latent variable $z_{t-1}$ for optimization. To align with this process, we define the energy function $\mathcal{E}(z_t, c, t) = \|z_{t-1} - z_{t-1}^*\|_1$ to condition the noise map for each timestep. Note that unlike NTI, we employ the L1 distance to compute the distance between the noise map and the current latent variable. Based on the definition of the energy function, Eq. 8 is revised as:

$$\epsilon_\theta(z_t, c_N) = -\sqrt{1-\alpha_t}(\nabla_{z_t} \log p(z_t) - s_g \cdot \nabla_{z_t}\|z'_{t-1} - z_{t-1}^*\|_1) \tag{9}$$

$$\text{where} \quad z'_{t-1} = \sqrt{\tfrac{\alpha_{t-1}}{\alpha_t}}z_t + \sqrt{\alpha_{t-1}}\left(\sqrt{\tfrac{1}{\alpha_{t-1}}-1} - \sqrt{\tfrac{1}{\alpha_t}-1}\right)\epsilon_\theta(z_t, \emptyset).$$

We term the noise map $c_N$ conditioned to produce the network output as $\epsilon_\theta(z_t, c_N)$ in Eq. 9. Additionally, we introduce a hyperparameter $s_g$ termed the gradient scale in Eq. 9 to compensate for the scale difference between the gradient and the network's output. By adjusting the magnitude of $s_g$, we modulate the degree of the edit from the original reverse process to the DDIM inversion trajectory. Building on the findings that a substantial guidance scale can yield high-quality results (Dhariwal & Nichol, 2021; Ho & Salimans, 2021), we also introduce a guidance technique for noise map condition similar to Eq. 2 as follows:

$$\tilde{\epsilon}_\theta(z_t, c_N) = \epsilon_\theta(z_t, \emptyset) + s_N \cdot (\epsilon_\theta(z_t, c_N) - \epsilon_\theta(z_t, \emptyset)), \tag{10}$$

where $s_N$ is the guidance scale of the noise map. To derive a latent variable conditioned on the noise map, we execute a one-step reverse process with $\tilde{\epsilon}_\theta(z_t, c_N)$ as follows:

$$z_{t-1}^{NM} = \sqrt{\tfrac{\alpha_{t-1}}{\alpha_t}}z_t + \sqrt{\alpha_{t-1}}\left(\sqrt{\tfrac{1}{\alpha_{t-1}}-1} - \sqrt{\tfrac{1}{\alpha_t}-1}\right)\tilde{\epsilon}_\theta(z_t, c_N). \tag{11}$$

By performing a text-conditioned reverse process with $z_{t-1}^{NM}$, a latent variable conditioned on a noise map, we derive our final latent variable that is conditioned on both the noise map and the text condition. Empirically, we find that we can approximate $z_t^{NM} \approx z_{t-1}^{NM}$. Thus, a latent $z_{t-1}$ both conditioned on noise map and text embedding is determined as follows:

$$\tilde{\epsilon}_\theta(z_t^{NM}, c_T) = \epsilon_\theta(z_t^{NM}, \emptyset) + s_T \cdot (\epsilon_\theta(z_t^{NM}, c_T) - \epsilon_\theta(z_t^{NM}, \emptyset)) \tag{12}$$

$$z_{t-1} = \sqrt{\tfrac{\alpha_{t-1}}{\alpha_t}}z_t^{NM} + \sqrt{\alpha_{t-1}}\left(\sqrt{\tfrac{1}{\alpha_{t-1}}-1} - \sqrt{\tfrac{1}{\alpha_t}-1}\right)\tilde{\epsilon}_\theta(z_t^{NM}, c_T). \tag{13}$$

To maintain consistent notation conventions with $s_N$ in Eq. 10, we designate $s_T$ to represent the text guidance scale, instead of $w$ in Eq. 2. In Figure 2 (c), we display the overall process of NMG. We note that NMG is a sequential process of first conditioning the noise map and conditioning the text embedding on the outcome of the step before.

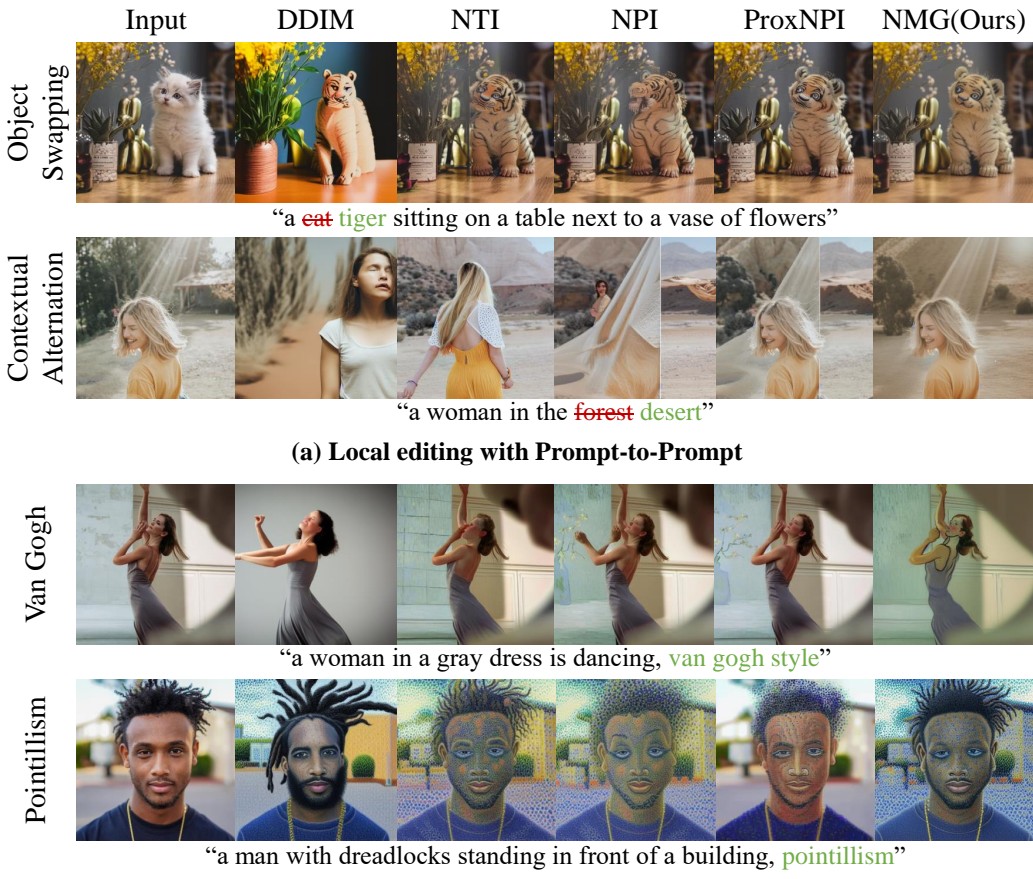

Input  DDIM  NTI  NPI  ProxNPI  NMG(Ours)

Object Swapping

"a ~~cat~~ tiger sitting on a table next to a vase of flowers"

Contextual Alternation

"a woman in the ~~forest~~ desert"

**(a) Local editing with Prompt-to-Prompt**

Van Gogh

"a woman in a gray dress is dancing, van gogh style"

Pointillism

"a man with dreadlocks standing in front of a building, pointillism"

**(b) Global editing with Prompt-to-Prompt**

Figure 3: Image editing results using Prompt-to-Prompt are shown in (a) for local editing and (b) for global editing. Results show that DDIM lacks in preserving details of the input image, both NTI and NPI face challenges in maintaining spatial context, and ProxNPI exhibits limited editing capabilities. In contrast, NMG consistently produces robust results for both local and global edits.

## 4 EXPERIMENTS

Our method is tailored to ensure that the reconstruction path remains closely aligned with the inversion trajectory throughout the image editing process. In the section below, we compare NMG with several methods, including DDIM (Song et al., 2020a), NTI (Mokady et al., 2023), NPI (Miyake et al., 2023), and ProxNPI (Han et al., 2023). While DDIM is not inherently formulated to prevent reconstruction divergence, we include it in our comparisons to highlight how image quality deteriorates when the reconstruction path strays from the desired trajectory.

### 4.1 QUALITATIVE COMPARISON

Given that the primary goal of inversion methods is image editing, we first integrate NMG into the widely adopted Prompt-to-Prompt (Hertz et al., 2022) editing method to empirically validate our approach. For the integration with Prompt-to-Prompt, we incorporate NMG within the reconstruction path, rather than the editing path. This approach stems from NMG's characteristic of aligning a pathway with the inversion trajectory. Such an alignment can be detrimental to the image's editing.

Prompt-to-Prompt performs diverse tasks by controlling attention maps from reconstruction and editing paths. Our experiments encompass four local editing tasks: object swapping, contextual alteration, facial attribute editing, and color change, as well as four global style transfer tasks to styles of oil paintings, van Gogh, pointillism, and watercolor. Figures 3 depict our results for both local and global editing using Prompt-to-Prompt. Due to its limited reconstruction capability, DDIM struggles to integrate details of the input image into the edited image. Both NTI and NPI leverage null-text embeddings to retain input image details, but since the null-text embedding is a one-dimensional

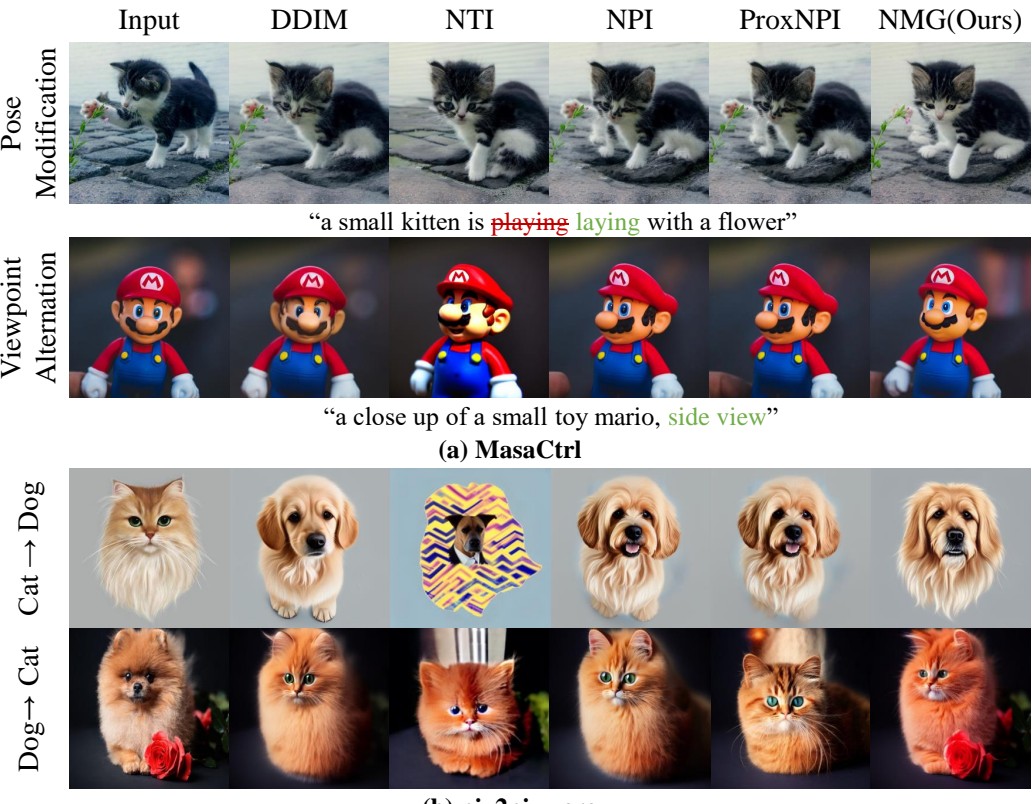

Figure 4: Image editing outcomes are presented using (a) MasaCtrl and (b) pix2pix-zero. NMG's proficiency in retaining spatial context is highlighted in (a), while its resilience to variations of DDIM inversion is showcased in (b).

vector, it inherently struggles to preserve spatial context. ProxNPI, due to the inversion guidance, utilizes the noise maps to follow an inversion trajectory and excels in retaining spatial context. However, this guidance is also applied to editing paths, producing constrained results. For instance, the first row of Figure 3 (b) illustrates an attempt to transition the style of the image toward "van Gogh style". ProxNPI tends to adhere closely to the original context, whereas our proposed NMG effectively transforms the style.

## 4.2 EVALUATING PROPERTY WITH ADDITIONAL EDITING METHODS

**Leveraging spatial context** The distinct advantage of NMG lies in its direct utilization of the noise maps, allowing it to preserve spatial context compared to other inversion methods. To demonstrate this capability, we integrate NMG with MasaCtrl (Cao et al., 2023), an editing approach that utilizes mutual self-attention for non-rigid editing tasks. Because MasaCtrl relies on the DDIM to reconstruct an input image during real image editing, we substitute the DDIM with NMG and other comparison methods to demonstrate the effectiveness of our approach. For spatially demanding edits, we undertake pose modification and viewpoint alternation. Figure 4 (a) showcases the results achieved with MasaCtrl. Due to NMG's capability to access spatial context directly through noise maps, it yields unparalleled editing results. The effectiveness of our method in preserving spatial context is further highlighted in Section 4.1. Although ProxNPI employs inversion guidance to capture spatial context, its reliance on a single gradient descent step to align with the inversion trajectory occasionally results in inconsistent outputs. We also compare results with ProxMasaCtrl, a method proposed by Han et al. (2023) that integrates ProxNPI with MasaCtrl using a different integration approach than our main comparison. For details and experiments, see Appendix A.2.

**Robustness to variations of DDIM inversion** Most inversion techniques leverage DDIM inversion as its foundation to encode the image into the latent. However, DDIM inversion can be modified to meet certain goals. To explore the robustness of our method to variations on DDIM inversion, we incorporate NMG with pix2pix-zero (Parmar et al., 2023). Pix2pix-zero, designed for zero-shot image-to-image translation, modifies DDIM inversion by adding a regularization term to calibrate the

| Models | P2P (Local) | | P2P (Global) | | MasaCtrl | | User Study |
|---|---|---|---|---|---|---|---|
| | CLIP↑ | TIFA↑ | CLIP↑ | TIFA↑ | CLIP↑ | TIFA↑ | |
| DDIM | 0.2977 | 0.8436 | 0.3066 | 0.8349 | 0.2825 | 0.8253 | - |
| NTI | 0.2983 | **0.9125** | 0.3202 | 0.8302 | 0.2903 | 0.8277 | 10.0% |
| NPI | 0.2982 | 0.9076 | 0.3157 | 0.8117 | 0.2921 | 0.8188 | 5.0% |
| ProxNPI | 0.2951 | 0.8947 | 0.3006 | 0.8463 | 0.2922 | 0.8188 | 12.5% |
| **NMG(Ours)** | **0.3007** | 0.8955 | **0.3221** | **0.8991** | **0.2955** | **0.8548** | **72.5%** |

Table 1: Quantitative evaluation of image editing using local and global editing with Prompt-to-Prompt, MasaCtrl, and user study reveals that NMG consistently surpasses other baseline methods in editing performance.

| | Method | | Reconstruction | | | |
|---|---|---|---|---|---|---|
| | optimized | conditional | MSE↓ | SSIM↑ | LPIPS↓ | Time↓ |
| NTI | ✓ | | 0.0127 | 0.7292 | **0.1588** | 104.85 |
| NMG | | ✓ | **0.0124** | 0.7296 | 0.1673 | **5.97** |
| NTI + NMG | ✓ | ✓ | **0.0124** | **0.7302** | 0.1668 | 107.72 |

Table 2: A quantitative evaluation of reconstruction assesses how closely each method aligns with the inversion trajectory. While combining both strategies yields the best results, NMG stands out as an efficient approach when considering both time and performance.

initial noise to more closely resemble Gaussian noise. In the inversion phase, we utilize the modified DDIM inversion proposed in pix2pix-zero. For the reconstruction phase, similar to our experiments with MasaCtrl, we integrate NMG and other comparison methods. Figure 4 (b) presents the editing outcomes achieved using pix2pix-zero on tasks like cat-to-dog, and dog-to-cat conversions. The results indicate that our method produces robust results with spatially coherent samples, even under variations of DDIM inversion.

## 4.3 QUANTITATIVE COMPARISON

To quantitatively measure the quality of image editing paired with our method, we utilize two metrics: CLIPScore (Hessel et al., 2021) and TIFA (Hu et al., 2023). CLIPScore gauges how closely the edited image aligns with the target text prompt in the CLIP (Radford et al., 2021) embedding space. TIFA, on the other hand, evaluates the semantic alignment of given image and text prompt pairs based on visual question-answering accuracy. Our evaluation, utilizing all the tasks described in Section 4.1, is conducted on four local and global editing results via Prompt-to-Prompt and two tasks for non-rigid image editing via MasaCtrl. We edit 20 images for each task and report the averaging scores in Table 1. It can be seen that NMG consistently surpasses competing methods, an efficacy that stems from NMG's capability to retain the spatial context of the input image without imposing constraints on the editing path.

While the metrics previously discussed are commonly employed, their reliance on specific models often causes them to misalign with human perception. To address this and evaluate visual quality from a human-centric perspective, we additionally conduct a user study. Fifty participants were recruited through *Prolific* (Prolific, 2023). For the study, 40 sets, each containing four images edited using NTI, NPI, ProxNPI, and NMG, were presented to the participants. Each set was paired with a specific editing instruction and an input image. Participants chose the image with the highest fidelity that best met the editing instructions. We note that images within each set were displayed in a randomized order. In our analysis, we collect the method most often selected in each comparison and report the selection ratio in the final column of Table 1. Notably, our NMG method was the most preferred by participants. These findings underscore the efficacy of our method in aligning closely with human perception of image quality. A print screen of the user study is provided in Figure 12.

## 4.4 ABLATION STUDY

**Reconstruction** For reconstruction of real images, NMG adheres to the trajectory of DDIM inversion, conditioned by noise maps. To demonstrate the effectiveness of our approach, we compare it with the optimization-based method, NTI. Our evaluation utilizes the MS-COCO validation set Lin et al. (2014), from which we randomly select 100 images. We assess the reconstruction quality of images using metrics such as MSE, SSIM (Wang et al., 2004), and LPIPS (Zhang et al., 2018). The

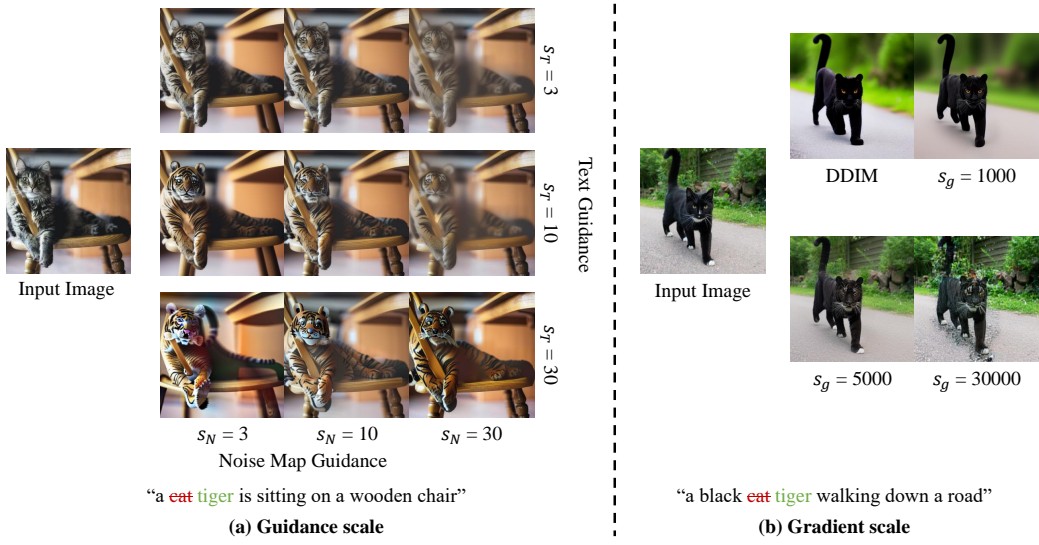

Figure 5: Ablation results of (a) guidance scales and (b) gradient scales. In (a), we demonstrate that the noise map guidance scale governs the influence of input image nuances, while the text guidance scale steers the extent of edits in the desired direction. In (b), we demonstrate that the gradient scale regulates the degree of alignment with the inversion trajectory.

results in Table 2 indicate that our method performs comparably to the optimization-based approach. Notably, the best results are achieved when both methods are used together. However, regarding reconstruction speed, our strategy outpaces its optimization counterpart by nearly a factor of 20. Taking both speed and performance into account, NMG emerges as the efficient choice for aligning with the inversion trajectory during reconstruction.

**Guidance scale**   NMG utilizes a dual-conditioning strategy comprising noise maps and text prompts. To discern the effects of each condition, we modulate their respective scales of $s_N$ and $s_T$ in Eq.10 and Eq. 12. Figure 5 (a) illustrates the influence of each guidance scale. Prioritizing text conditioning aligns outcomes more closely to the intended edits. In contrast, a prominent noise map conditioning scale retains the image's context. However, achieving a balance between the two scales is essential for a desirable outcome. Over-relying on text conditioning, evident in the grid image's lower left triangle, erodes the nuance of the input image. Conversely, an excessive emphasis on noise map conditioning, seen in the grid image's upper right triangle, may hinder successful editing. The grid image's diagonal showcases results from balanced scaling, indicating that appropriate scaling yields the best outcomes. Importantly, our experiments maintained a consistent guidance scale, underscoring its robustness across varied samples.

**Gradient scale**   To regulate the adherence to the trajectory of DDIM inversion, we employ a gradient scale $s_g$ in Eq. 9. The effects of this scale are demonstrated by varying its magnitude and presenting the samples in Figure 5 (b). A smaller gradient scale offers limited alignment with the inversion trajectory, leading to edited outputs closely mirroring the result using DDIM in the reconstruction path. Conversely, an overly large gradient scale results in pronounced trajectory alignments, degrading the editing quality. Therefore, the proper selection of the gradient scale is vital. Analogous to the guidance scale, we maintain a consistent gradient scale across all experiments, ensuring it remains universally effective rather than overly sensitive to individual samples.

## 5   CONCLUSION

In the evolving field of image editing, Noise Map Guidance (NMG) offers a notable advancement. By addressing the challenges present in current real-image editing methods using text-guided diffusion models, NMG introduces an inversion method rich in spatial context. NMG directly conditions noise maps to the reverse process, which captures the spatial nuance of the input image and ensures the preservation of its spatial context. Experimental results demonstrate NMG's capability to preserve said spatial context, and this property is furthermore highlighted with spatially intensive edits. NMG is also designed as an optimization-free approach that prioritizes speed without compromising quality. NMG represents a significant step forward, suggesting further investigation and refinement in real-image editing techniques.

## ACKNOWLEDGEMENTS

We thank the ImageVision team of NAVER Cloud for their thoughtful advice and discussions. Training and experiments were done on the Naver Smart Machine Learning (NSML) platform (Kim et al., 2018). This study was supported by BK21 FOUR. T.-H. Oh was partially supported by Institute of Information & communications Technology Planning & Evaluation (IITP) grant funded by the Korea government(MSIT) (No.2021-0-02068, Artificial Intelligence Innovation Hub; No.RS-2023-00225630, Development of Artificial Intelligence for Text-based 3D Movie Generation).

## ETHICS STATEMENT

Generative models for synthesizing images carry with them several ethical concerns, and these concerns are shared by (or perhaps exacerbated in) any generative models such as ours. Generative models, in the hands of bad actors, could be abused to generate disinformation. Generative models such as ours may have the potential to displace creative workers via automation. That said, these tools may also enable growth and improve accessibility for the creative industry.

## REPRODUCIBILITY STATEMENT

The source code can be found at https://github.com/hansam95/NMG.

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

# Noise Map Guidance: Inversion with Spatial Context for Real Image Editing

## – Supplementary Material –

## A    ADDITIONAL EXPERIMENTS

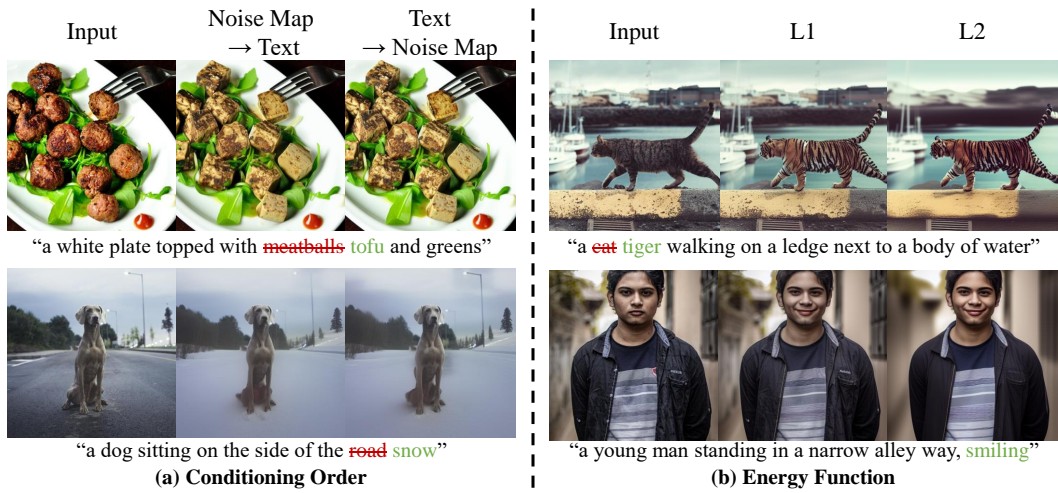

Figure 6: Ablation results of (a) conditioning order and (b) energy function

### A.1    ADDITIONAL ABLATION STUDY

**Conditioning Order**    As detailed in Section 3.2, NMG employs a dual-conditioning approach that leverages both the noise map and text conditions sequentially. While our primary strategy conditions the noise map before the text embeddings, it is feasible to reverse this order. Figure 6 (a) presents the outcomes of local editing under varying conditioning sequences: the second column depicts the canonical NMG sequence—initially the noise map, then text; the third column illustrates the reverse order. Both configurations yield promising results, underscoring NMG's robustness to conditioning sequence variations.

**Energy Function**    In Section 3.2, we formulate the energy function using the L1 distance, diverging from the L2 distance employed by NTI (Mokady et al., 2023). Figure 6 (b) showcases editing outcomes derived from these distinct energy functions: the second column illustrates results using L1 distance, while the third column represents those from L2 distance. Both methodologies adeptly guide the editing direction. However, the L2-based results manifest a noticeable blurring in the image background. These observations align with the findings of prior research (Isola et al., 2017; Pathak et al., 2016) that L2 distance often produces blurred results. Hence, for NMG, we intentionally select the L1 distance for energy function formulation.

### A.2    COMPARISONS WITH PROXMASACTRL

ProxiNPI (Han et al., 2023) introduces an integration with MasaCtrl (Cao et al., 2023), termed ProxMasaCtrl. This approach incorporates NPI (Miyake et al., 2023) in the reconstruction path, while leveraging proximal guidance in the editing path to enhance reliability. Figure 7 compares the results from ProxMasaCtrl with our integration, labeled as "NMG + MasaCtrl". Despite ProxMasaCtrl's mitigation of unintended changes through proximal guidance, its use of NPI in reconstruction means it often misses spatial context. This lack of spatial context is evident in Figure 7 (a): the first row highlights ProxMasaCtrl's inability to alter poses accurately, and the second row reveals missing spatial components, like a kitten's tail, during editing. Similarly, when observing viewpoint alternation results in Figure 7 (b), ProxMasaCtrl struggles to deliver convincing outcomes, whereas our NMG-integrated solution, drawing on its rich spatial context, consistently achieves superior results.

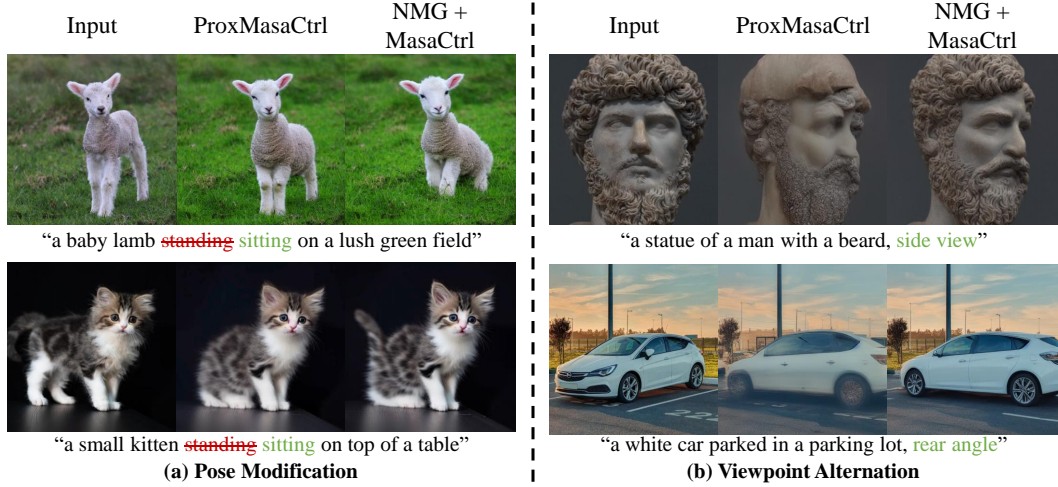

| Input | ProxMasaCtrl | NMG + MasaCtrl | | Input | ProxMasaCtrl | NMG + MasaCtrl |

"a baby lamb ~~standing~~ sitting on a lush green field"

"a statue of a man with a beard, side view"

"a small kitten ~~standing~~ sitting on top of a table"

"a white car parked in a parking lot, rear angle"

**(a) Pose Modification**

**(b) Viewpoint Alternation**

Figure 7: Comparison with ProxMasaCtrl

## A.3 EXPERIMENTS DETAILS

Within our experimental framework, we employ Stable Diffusion (Rombach et al., 2022), standardizing the diffusion steps to $T = 50$ across all experiments. For the editing tasks, the parameters are set as follows: noise map guidance $s_N = 10$, text guidance $s_T = 10$, and guidance scale $s_g = 5000$. For the reconstruction tasks, the configurations are set to $s_N = 10$, $s_T = 7.5$, and $s_g = 10000$.

## A.4 ADDITIONAL RESULTS

**Local and Global Editing**   We present supplementary comparison results in Figure 10 and Figure 11, showcasing local and global editing outcomes achieved with Prompt-to-Prompt (Hertz et al., 2022). These additional results further underscore NMG's proficiency in preserving spatial context without diminishing the quality of the editing output. Furthermore, we present the results of diverse stylization tasks in Figure 8(a), demonstrating NMG's versatility. This evidence suggests that NMG's capabilities extend beyond painting stylizations, encompassing a broad spectrum of styles, including anime and mosaic.

**Non-rigid Editing**   In conjunction with MasaCtrl (Cao et al., 2023), NMG demonstrates its capability for non-rigid editing. Figure 8(b) exhibits additional experimental results of pose modifications. It is evident that NMG effectively conducts edits with pronounced spatial information changes.

**Editing with NTI + NMG**   In image reconstruction, the combined use of NTI and NMG has been shown to yield superior quality. To ascertain whether this synergy also extends to image editing, we have undertaken a series of editing experiments utilizing both NTI and NMG. Figure 8(c) demonstrates that while NMG alone provides dependable editing, it occasionally lacks specific spatial details. Integrating additional information from NTI effectively compensates for this shortfall, restoring the spatial context of the input image. This integration potentially represents a significant advancement in augmenting the capabilities of NMG for image editing tasks.

**Comparisons with ProxNPI**   For a further comparative analysis with ProxNPI, we replicated the experiment using the same image featured in the ProxNPI study. Figure 8(d) presents these additional comparative results. The outcomes demonstrate that NMG yields results comparable to the ProxNPI. However, it is essential to note that ProxNPI primarily concentrates on object swapping and does not extensively explore various editing scenarios, such as contextual alterations or stylization. As discussed in Section 4.1, ProxNPI's capabilities are somewhat limited due to its reliance on inversion guidance. In contrast, NMG is not subject to these constraints, demonstrating its more extensive utility in various image editing tasks.

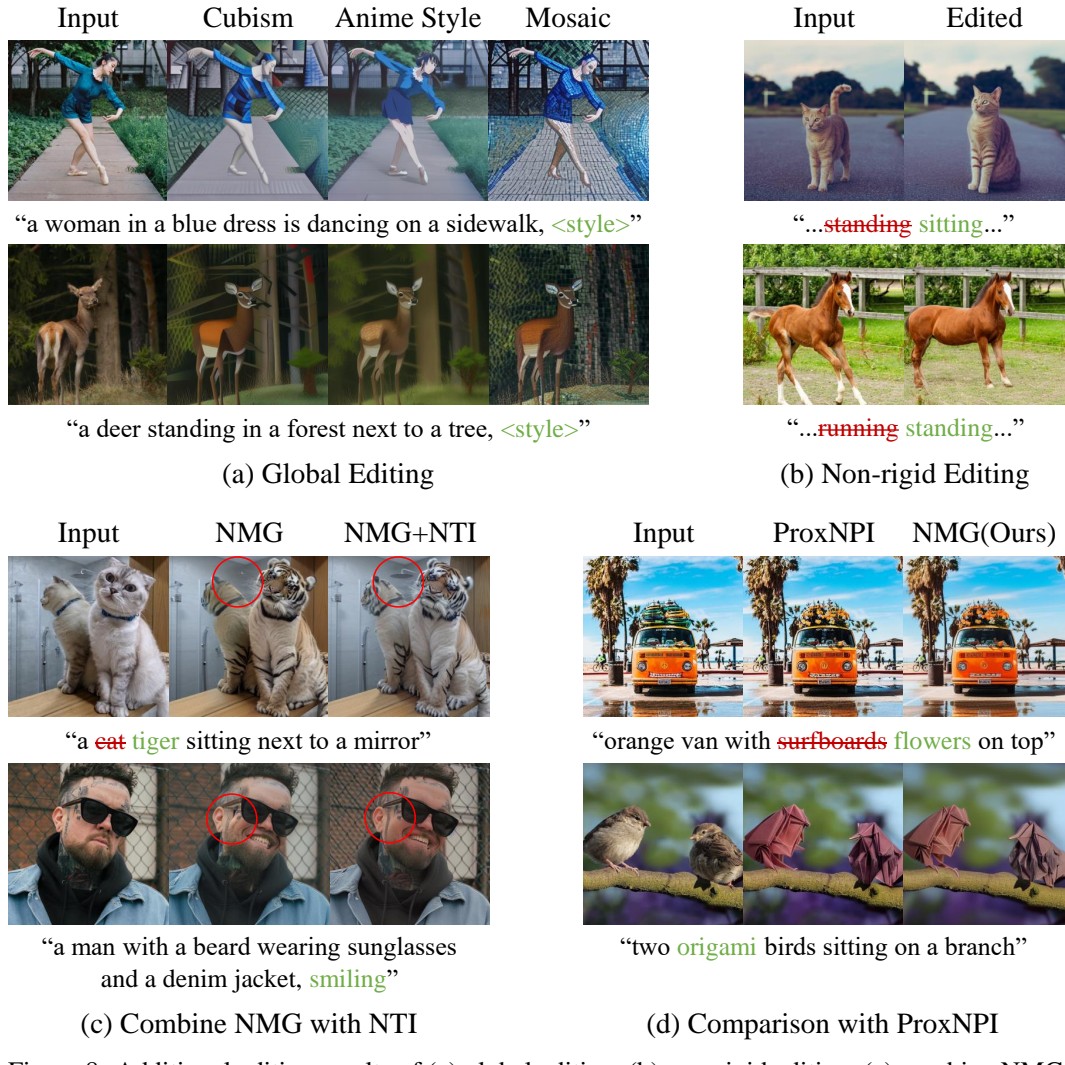

Figure 8: Additional editing results of (a) global editing, (b) non-rigid editing, (c) combine NMG with NTI, and (d) additional comparison with ProxNPI

**Reconstruction** We offer an additional quantitative comparison of reconstruction results. The left side of Table 3 presents a quantitative comparison of reconstruction capabilities between NMG and other methods. While NMG demonstrates comparable quantitative results in reconstruction to these methods, it is important to emphasize that NMG's primary focus is on image editing. It should be noted that high-quality reconstruction does not invariably equate to superior editing performance. ProxNPI exhibits a commendable ability to reconstruct the image in Table 3. However, as illustrated in the sixth row of Figure 11, its editing capabilities are somewhat limited, underscoring the distinction between reconstruction proficiency and editing versatility.

**Additional User Study** Evaluating edited images based on their alignment with editing instructions is crucial, yet it is equally essential to delve into more detailed aspects of evaluation. For example, in local editing tasks, it is essential to maintain the integrity of unedited regions, while in global editing, preserving the overall structure of the image is essential. To explore these aspects, we conducted an additional user study similar to the approach in Section 4.3. Thirty participants, recruited via *Prolific* (Prolific, 2023), were presented with ten sets of images. Participants were instructed to assess the preservation of unselected regions in local editing and the retention of the overall structure in global editing. The results of the study, displayed on the right side of Table 3, indicate that NMG performs well in both preserving unselected regions during local editing and maintaining the overall structure in global editing scenarios.

| Method | Reconstruction | | | User Study |
|---|---|---|---|---|
| | MSE↓ | SSIM↑ | LPIPS↓ | |
| DDIM | 0.159 | 0.368 | 0.521 | - |
| NTI | 0.013 | 0.729 | 0.159 | 0.0% |
| NPI | 0.021 | 0.661 | 0.230 | 0.0% |
| ProxNPI | 0.012 | 0.737 | 0.153 | 10.0% |
| **NMG(Ours)** | 0.012 | 0.723 | 0.167 | 90.0% |

Table 3: Quantitative comparison of reconstruction and additional user study

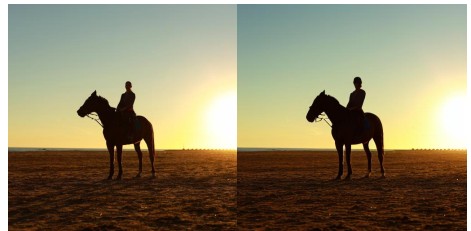

"a person ~~riding~~ beside a horse "

(a) Relationship Change

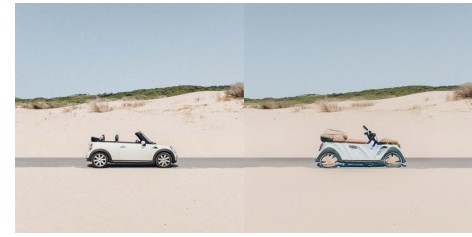

"a ~~car~~ bicycle driving down a sandy road"

(b) Object Swap with Different Structure

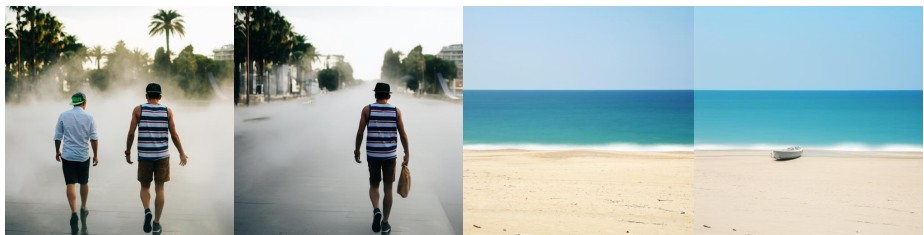

"~~two men~~ one man walking down a street"        "a boat on the right side of the beach"

(c) Object Adding and Deleting

Figure 9: Illustrations of limitations

## B  LIMITATIONS

Our research endeavors to enhance inversion-based editing methods, focusing on improving spatial context during the editing process. Nevertheless, our current methodology is limited to compatibility with methods that do not conform to the inversion-based framework. For instance, SGC-Net (Zhang et al., 2022), which operates on a text-guided diffusion model for relationship change tasks, proves challenging to integrate with NMG due to its deviation from the inversion-based editing paradigm. Consequently, applying NMG in conjunction with SGC-Net for tasks involving relationship changes faces significant hurdles. As an alternative, we attempted relationship change tasks using MasaC-trl (Cao et al., 2023). However, Figure 9(a) depicts the ineffectiveness of this approach, as MasaCtrl is not inherently designed for relationship change tasks.

Additionally, the editing capabilities of NMG are intrinsically linked to the limitations of existing inversion-based methods. For example, Prompt-to-Prompt (Hertz et al., 2022) editing involves the swapping of cross-attention maps between source and target texts, which restricts the possibility of structural changes in the edited object to the constraints of the original image's object. This limitation is evident in Figure 9(b), which illustrates an object swap task between a car and a bicycle. The disparate structures of these objects underscore the challenges faced by NMG in ensuring reliable editing under these conditions.

Moreover, NMG's dependence on text for image editing hinders its ability to perform precise spatial changes. As shown on the left side of Figure 9(c), NMG can effectively remove an object, such as a man, from an image. However, it lacks the precision to identify and remove a specific individual,

making targeted removals unfeasible. Similarly, as seen on the right side of Figure 9(c), NMG struggles to follow exact location directives when adding new objects. For instance, despite the instruction to place a boat on the right side of the beach, the edited image fails to place it on the right. Addressing these challenges to enable more accurate spatial control in real image editing with NMG is a vital area for future research and development.

## C    EXTENDED RELATED WORK

**Guidance in Diffusion Models**    To generate samples that abide by a certain condition, Score-SDE (Song et al., 2020b) initially introduces a formulation for conditional diffusion models. Following this formulation, Dhariwal & Nichol (2021) proposes classifier guidance, which leverages additional classifiers to enhance the quality of class conditional image synthesis. Building upon classifier guidance, Ho & Salimans (2021) proposes classifier-free guidance, which obviates the need for an additional classifier. However, these guidance methods are often restricted to class or text conditions. EGSDE (Zhao et al., 2022) advances the field by introducing energy guidance techniques. Although EGSDE primarily focuses on image-to-image translation tasks, they notably expand the flexibility in conditioning formats of diffusion models. Similarly, DiffuseIT (Kwon & Ye, 2022), inspired by MCG (Chung et al., 2022), focuses on image translation. Successive methodologies, such as FreeDoM (Yu et al., 2023) and universal guidance (Bansal et al., 2023), broaden the scope of conditions to encompass a diverse range of generative tasks. These methodologies are adaptable to various conditional signals, including segmentation maps, style images, and facial identities. Aligning with these advancements, our approach also leverages the versatile framework of energy guidance, targeting the accurate reconstruction of input images.

## D    USER STUDY

A screenshot of our user study conducted through Prolific (Prolific, 2023) is depicted in Figure 12. Note that Prolific receives a pre-built Google form, and distributes it to participants of the study.

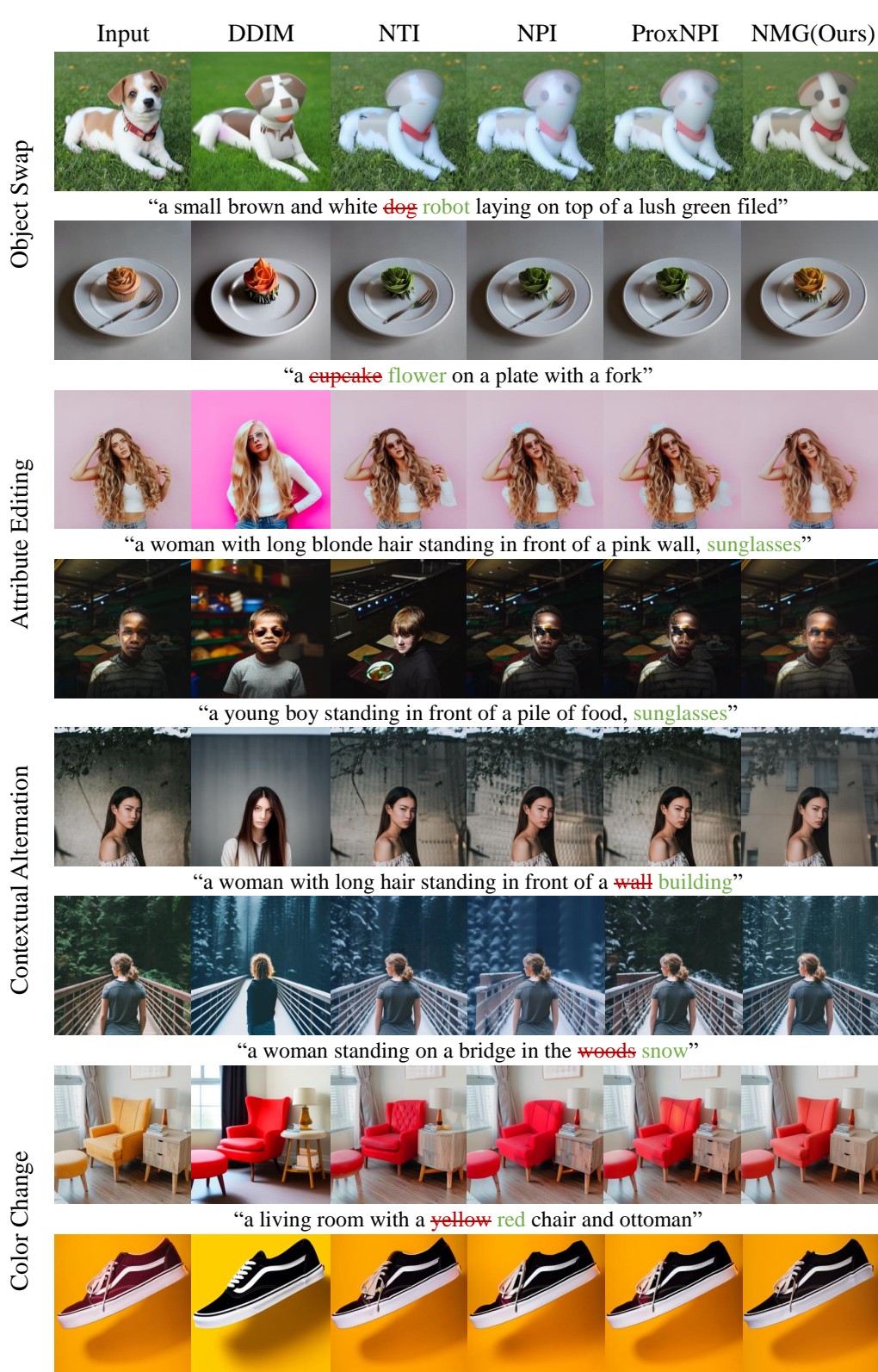

Figure 10: Local editing with Prompt-to-Prompt

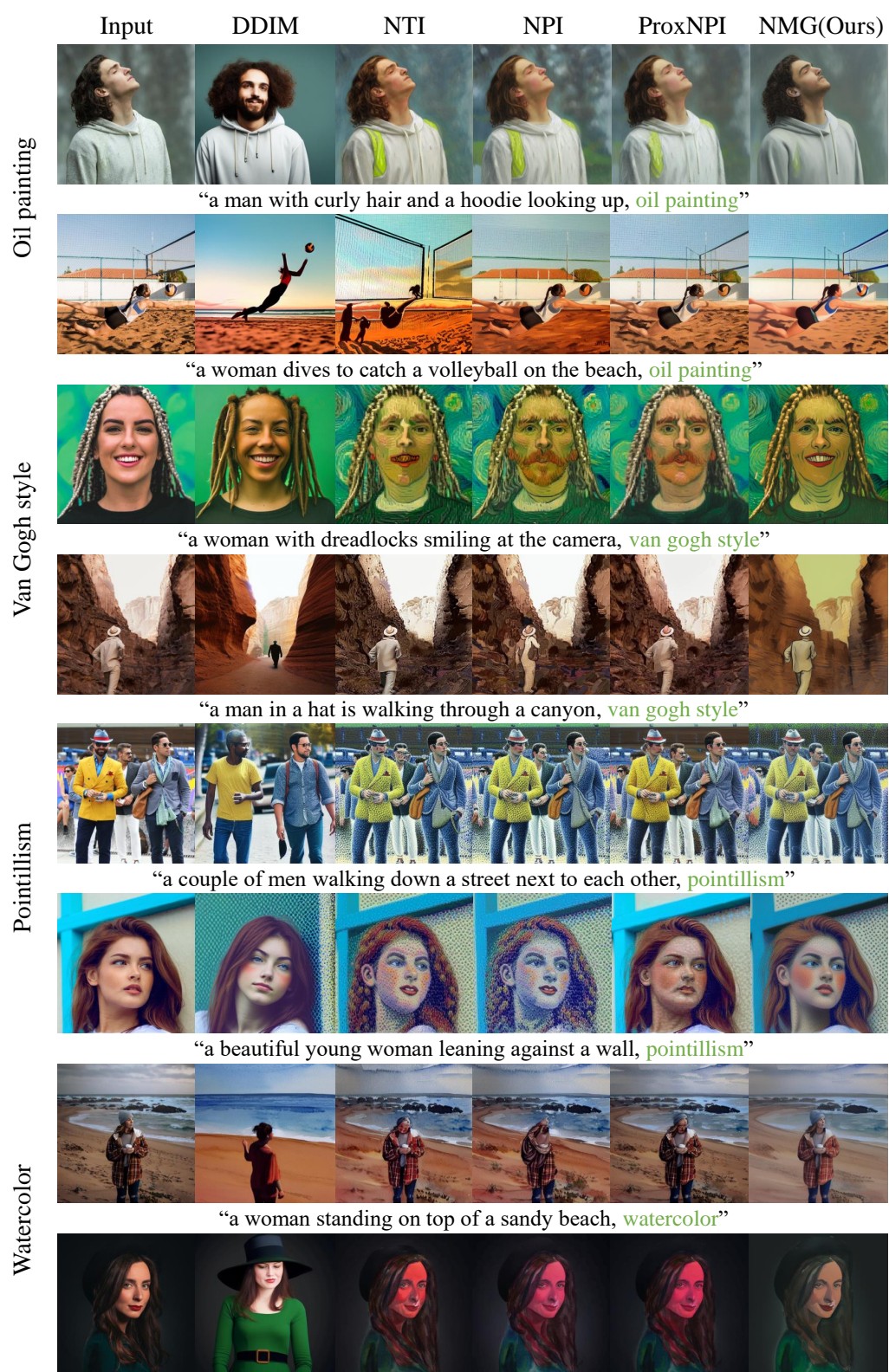

Figure 11: Global editing with Prompt-to-Prompt

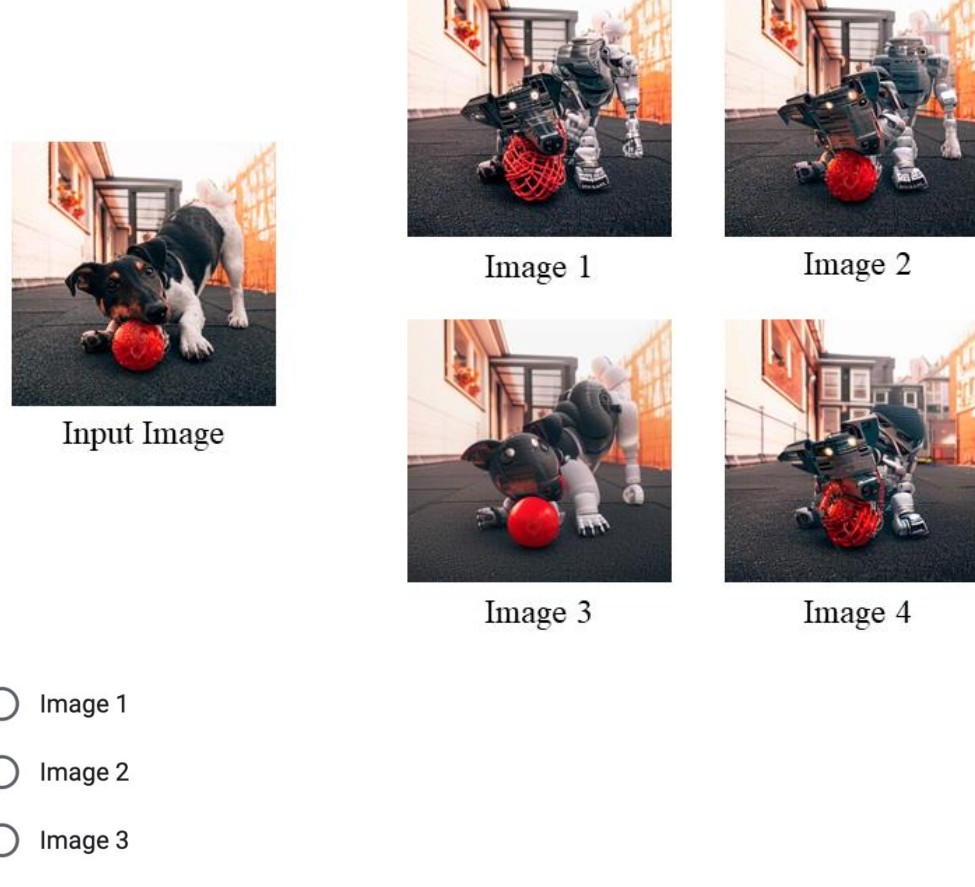

Figure 12: User study screenshot

