# OpenReview forum: "Noise Map Guidance: Inversion with Spatial Context for Real Image Editing"
_ICLR.cc/2024/Conference — ICLR 2024 poster_

### Official Review · Reviewer_pVhT · 2023-10-30

**Soundness:** 3 good
**Presentation:** 3 good
**Contribution:** 3 good
**Rating:** 6
**Confidence:** 3

**Summary:**

The paper "Noise Map Guidance: Inversion with Spatial Context for Real Image Editing" presents a novel inversion method called Noise Map Guidance (NMG) for real-image editing using text-guided diffusion models. NMG addresses the challenges faced by existing methods, such as Null-text Inversion (NTI), which fail to capture spatial context and require computationally intensive per-timestep optimization. NMG achieves high-quality editing without necessitating optimization by directly conditioning noise maps to the reverse process, capturing the spatial context of the input image. The authors demonstrate NMG's adaptability across various editing techniques and its robustness to variants of DDIM inversions through empirical investigations

**Strengths:**

This paper is well-written and provides sufficient background and analysis into the motivations and effectiveness of NMG. the overall framework is very straightforward, there should be no difficulty for other to reproduce. it demonstrates a strong adaptability across various editing techniques, including Prompt-to-Prompt, MasaCtrl, and pix2pix-zero.

The method is optimization-free, making it computationally efficient while preserving editing quality, it achieve a 20 times acceleration compares to null-text inversion.

comprehensive quantitative and qualitative comparison of NMG with other inversion methods, showcasing its superior performance in preserving spatial context and editing fidelity.

**Weaknesses:**

NMG impose a very strong spatial constraint during editing, as in the figure most showcase have almost the same geometry structure as the original picture, for case the modify geometry e.g. the cat in figure 4, the result shows an apparent artifacts in the modified region. There needs a further investigation on how will NMG perform when facing editing that requires modification on the spatial structure, for example removing a target (like "two man ..." ->"one man...") or change to a totally different object ("...car" to "... bike " ).

Moreover, there lack of discussion about possible failure cases of  NMG, the authors should add such discussion about in what circumstance NMG would fail and the reason why it fails to help the community better understand the proposed method.

**Questions:**

Why there are no quantitive and qualitative comparison with previous works about reconstruction? I think there should be a comparison with other methods in this aspect, or the author should explain why it is omitted.

How NTI + NMG  performs when dealing with actual editing task? it would be helpful to show the proposed method can combine with previous method to achieve a better result.

---

> ### Author Response · Authors · 2023-11-21
> **Response to Reviewer pVhT**
>
> > **[W1]** NMG impose a very strong spatial constraint during editing, as in the figure most showcase have almost the same geometry structure as the original picture, for case the modify geometry e.g. the cat in figure 4, the result shows an apparent artifacts in the modified region. There needs a further investigation on how will NMG perform when facing editing that requires modification on the spatial structure, for example removing a target (like "two man ..." ->"one man...") or change to a totally different object ("...car" to "... bike " ). Moreover, there lack of discussion about possible failure cases of NMG, the authors should add such discussion about in what circumstance NMG would fail and the reason why it fails to help the community better understand the proposed method.
>
> We have newly appended additional editing results that modify the spatial structure of the object in Figure 9(b) and (c) in the revised paper. Figure 9(b) underscores NMG's challenges in tasks that demand significant object structural changes. This is partly due to the inherent limitations of the inversion-based methods upon which NMG's editing capabilities are built. For instance, in Prompt-to-Prompt editing, the method involves swapping cross-attention maps between source and target texts. This approach inherently limits the scope of structural changes to those permissible within the constraints of the original object’s geometry. Additionally, as shown in Figure 9(c), we encounter limitations when removing specific targets. While NMG can eliminate an object from an image, the method often lacks the precision needed for detailed locational choices, making it challenging to select and remove specific targets. These examples highlight areas where NMG, in its current form, struggles to perform edits that require detailed spatial adjustments or significant geometric transformations. We have added this discussion in Section B of the revised paper as a limitation discussion.
>
> >**[Q1]** Why there are no quantitive and qualitative comparison with previous works about reconstruction? I think there should be a comparison with other methods in this aspect, or the author should explain why it is omitted.
>
> Thank you for the question regarding the absence of comparisons with previous works, specifically in the context of reconstruction. We newly appended an additional quantitative comparison of reconstruction in Table 3 in the Appendix of the revised paper. NMG shows comparable performance with the comparison method in reconstruction. It's important to emphasize that NMG focuses primarily on image editing rather than reconstruction. This distinction is pivotal as we aim to advance the field of image editing, where the quality of editing does not necessarily correlate directly with the quality of reconstruction. For instance, as shown in the sixth row of Figure 11, while ProxNPI exhibits commendable reconstruction ability, its editing capabilities are somewhat limited, highlighting the difference between reconstruction proficiency and editing versatility. We have added this discussion in Section A.4 in the Appendix of the revised paper.
>
> >**[Q2]** How NTI + NMG performs when dealing with actual editing task? it would be helpful to show the proposed method can combine with previous method to achieve a better result.
>
> We have conducted additional experiments to explore how the synergy between NTI and NMG translates to actual image editing tasks. As shown in Figure 8(c), we observed that while NMG alone is a reliable tool for editing, it sometimes falls short in capturing specific spatial details. However, when we integrate additional information from NTI with NMG, this shortfall is effectively compensated for, enhancing the spatial context preservation in the edited images. This integration of NTI and NMG has shown promising results in our experiments, indicating a significant improvement in the editing capabilities of NMG. Thanks for the suggestion, which indeed improves our work further.

---

> ### Author Response · Authors · 2023-11-23
> **Looking forward to your response**
>
> Dear Reviewer pVhT,
>
> We are grateful for your in-depth and insightful evaluation. As the end of the rebuttal period draws near, we are making an effort to ascertain if there are any outstanding questions or matters that require clarification. Recognizing the significant time pressures you are under and the importance of meticulous review, we sincerely encourage any additional comments or insights you might have regarding our response. Your time and meaningful involvement in the peer review process are deeply appreciated.

---

### Official Review · Reviewer_9KmB · 2023-10-31

**Soundness:** 3 good
**Presentation:** 3 good
**Contribution:** 3 good
**Rating:** 8
**Confidence:** 4

**Summary:**

This paper proposes a noise map guidance method to capture the spatial context information in the input image, addressing the challenge in Null-text Inversion (NTI).  The proposed method is designed for real image editing. Experiments are performed to demonstrate various image editing capabilities of NMG, such as face expression modification, style transfer, viewpoint alternation, among others.

**Strengths:**

The proposed method looks simple but effective;
Both quantitative and qualitative experiments are performed to validate that NMG outperforms baselines;
Extensive experiments are performed to validate that NMG is able to achieve different image editing tasks.

**Weaknesses:**

I have the following comments about the weaknesses:

1) Figure 2 seems confusing to me. Since this paper repeatedly mentioned that NMG address the challenges in Null-text Inversion, I think it would be nice to compare NMG to Null-text Inversion in this Figure, and demonstrate how NMG outperforms Null-text Inversion. In addition, it would be nice to add the corresponding text descriptions in the Figure. *E.g.*, it's unclear whether the caption indicate to change the blue fire hydrant to a red one or it's just some deviations during the reconstruction.

2) Limitations and future work should be discussed. For example, whether NMG can address the relationship change task like SGC-Net [1]? Whether NMG can achieve various non-rigid image editing tasks like Imagic [2]? It seems that NMG can achieve some non-rigid editing tasks such as viewpoint alternation or face expression modification. However, spatial information between the output and input seems consistent in the majority parts, from my view. Thus, it would be great to see experiments exploring whether NMG can perform other operations (with more obvious spatial information change) such as "from a tiger" to "a jumping/running tiger".

[1] Zhang, Zhongping, et al. Complex Scene Image Editing by Scene Graph Comprehension. BMVC 2023.
[2] Kawar, Bahjat, et al. Imagic: Text-based real image editing with diffusion models. CVPR 2023.

**Questions:**

See Weaknesses.

---

> ### Author Response · Authors · 2023-11-21
> **Response to Reviewer 9KmB**
>
> >**[W1]** Figure 2 seems confusing to me. Since this paper repeatedly mentioned that NMG address the challenges in Null-text Inversion, I think it would be nice to compare NMG to Null-text Inversion in this Figure, and demonstrate how NMG outperforms Null-text Inversion. In addition, it would be nice to add the corresponding text descriptions in the Figure. E.g., it's unclear whether the caption indicate to change the blue fire hydrant to a red one or it's just some deviations during the reconstruction.
>
> Thank you for insightful feedback on Figure 2. We acknowledge the necessity of clearly demonstrating how our NMG approach effectively addresses the challenges associated with NTI. In response to suggestions, we have revised Figure 2 for enhanced clarity. In the updated Figure 2(a), we illustrate the common divergence issues encountered in naive reconstruction with an unaligned reverse process. This part of the figure highlights how such reconstructions often fail due to a deviation from the original inversion path, necessitating an alignment to achieve reliable reconstruction, which motivates our work. Figure 2(b) then delves into the alignment process via NTI, which optimizes the null-text embedding to reduce the disparity between the inversion and reconstruction paths. However, NTI's reliance on a single vector of the null-text embedding for alignment lead to challenges in preserving the spatial context of the input image due to having no spatial dimension of the null-text embedding. In contrast, as shown in Figure 2(c), our NMG offers a novel approach to alignment. Our method corrects the reconstruction path by conditioning it on the discrepancies between the inversion and reconstruction paths. NMG utilizes a noise map for this process, which inherently contains a richer spatial context than that of the null text embedding. This direct use of the noise map in NMG allows for more effective preservation of the spatial context during the editing process.
>
> >**[W2]** Limitations and future work should be discussed. For example, whether NMG can address the relationship change task like SGC-Net [1]? Whether NMG can achieve various non-rigid image editing tasks like Imagic [2]? It seems that NMG can achieve some non-rigid editing tasks such as viewpoint alternation or face expression modification. However, spatial information between the output and input>t seems consistent in the majority parts, from my view. Thus, it would be great to see experiments exploring whether NMG can perform other operations (with more obvious spatial information change) such as "from a tiger" to "a jumping/running tiger".
>
> Thank you for the question regarding the capabilities of NMG. We acknowledge the importance of discussing our work's limitations and potential future directions.
>
> **[W2-1]**
> We have appended the limitation section in Section B, Appendix of the revised paper. As detailed in Section B, our current methodology is primarily designed to enhance inversion-based editing methods, focusing on improving spatial context during the editing process. However, NMG faces challenges integrating with methods like SGC-Net, which operates on a text-guided diffusion model for relationship change tasks; thus, not compatible with our current method. This is due to SGC-Net's deviation from the inversion-based editing paradigm, and it is non-trivial to design the integration. As an alternative, we attempted relationship change tasks using MasaCtrl in Figure 9(a) in the  Appendix of the revised paper. However, Figure 9(a) depicts the ineffectiveness of this approach, as MasaCtrl is not inherently designed for relationship change tasks. Consequently, NMG fails to conduct relationship change tasks.
>
> **[W2-2]**
> We have conducted additional edited results of the non-rigid image editing task in Figure 8(b) in the revised paper. Figure 8(b) shows that NMG effectively conducts edits with pronounced spatial information changes, like standing to sitting and running to standing.

---

> ### Author Response · Authors · 2023-11-23
> **Looking forward to your response**
>
> Dear Reviewer 9KmB,
>
> Thank you for your comprehensive and perceptive review. With the close of the rebuttal period approaching, we're reaching out to check if there are any additional queries or issues that need addressing. Acknowledging the substantial time constraints you face and the value of detailed consideration, we warmly invite your further thoughts and feedback on our response. We are grateful for the time you've invested and your valuable participation in the peer review process.

---

### Official Review · Reviewer_yyX5 · 2023-10-31

**Soundness:** 4 excellent
**Presentation:** 4 excellent
**Contribution:** 3 good
**Rating:** 6
**Confidence:** 4

**Summary:**

The paper introduces NOISE MAP GUIDANCE (NMG), a new method for real-image editing. NMG uses noise maps derived from latent variables of DDIM inversion to capture spatial context effectively. By conditioning these noise maps to the reverse process and using both noise maps and text embeddings for image editing, NMG eliminates the need for time-consuming optimization. The results show that NMG preserves spatial context better, works faster, and integrates well with various other editing techniques while maintaining high edit quality. Furthermore, it demonstrates robust performance across different versions of DDIM inversion.

**Strengths:**

- The presented concept is intriguing and efficient. It details an uncomplicated yet effective method of using noise map conditioning during real image inversion, which streamlines the reverse process and eradicates path divergence between the reconstruction path and inversion trajectory. This leads to a more precise reconstruction.

- Moreover, the experiments carried out are robust. They demonstrate superior performance in real-image editing both qualitatively and quantitatively. Additionally, the tests focusing on spatial context utilization are crucial, effectively proving enhanced spatial context preservation capabilities.

**Weaknesses:**

- The visualization results depict many recurring stylization outcomes, such as the "oil painting style" and "Van Gogh style". It would be beneficial for the paper to exhibit a broader variety of more challenging editing instances.

- The user study could benefit from providing more accurate directives in its questions; if it pertains to local editing, for instance, the question should contemplate including "evaluate original preservation in unedited areas”.

**Questions:**

- How about the editing performance of adding or deleting elements in the images?

- While both global and local editing in the ProxNPI paper seem promising, the editing capability in this paper doesn't appear as effective. For instance, in Figure 3, the second row shows an edited result with a clear boundary between two types of backgrounds. In the third row, under "Van Gogh," the overall style seems to have undergone minimal change. Can you provide an explanation for these observations?

---

> ### Author Response · Authors · 2023-11-21
> **Response to Reviewer yyX5**
>
> > **[W1]** The visualization results depict many recurring stylization outcomes, such as the "oil painting style" and "Van Gogh style". It would be beneficial for the paper to exhibit a broader variety of more challenging editing instances.
>
> We have newly appended additional stylization results in Figure 8(a) and a description to Section A.4 in the Appendix of the revised paper. In this figure, we present additional results that showcase NMG's capabilities in handling a wide range of artistic styles, extending well beyond the initially noted painting stylizations. Specifically, the figure includes examples of editing outcomes in styles like anime and mosaic. This diversity in style adaptation illustrates the flexibility and robustness of NMG in accommodating various artistic expressions and editing challenges.
>
> >**[W2]** The user study could benefit from providing more accurate directives in its questions; if it pertains to local editing, for instance, the question should contemplate including "evaluate original preservation in unedited areas.”
>
> We agree that a precise evaluation framework is crucial for assessing the performance of our method. To this end, we have conducted an additional user study and have detailed the methodology and results in Section A.4 and Table 3 in the Appendix of the revised paper. This study is designed with a particular focus on evaluating the preservation of integrity in unedited areas during local editing tasks and the retention of overall structure in global editing scenarios. As evidenced in Table 3, the outcomes affirm that NMG successfully maintains unedited regions during local editing tasks and preserves the overarching structure in global edits, underlining its effectiveness in diverse editing contexts.
>
> >**[Q1]** How about the editing performance of adding or deleting elements in the images?
>
> We have added additional results in Figure 9(c) in the revised paper. NMG can eliminate objects from an image, such as removing a person. However, it is essential to note that while NMG can successfully execute such deletions, it is currently limited in its specificity when selecting individual targets for removal. Similarly, NMG faces challenges in adhering to detailed locational instructions when adding new objects to an image. Although NMG provides reliable editing for general tasks, its reliance on text for image editing introduces difficulties in executing detailed spatial changes. We have added this discussion in Section B of the revised paper as a limitation discussion.
>
> >**[Q2]** While both global and local editing in the ProxNPI paper seem promising, the editing capability in this paper doesn't appear as effective. For instance, in Figure 3, the second row shows an edited result with a clear boundary between two types of backgrounds. In the third row, under "Van Gogh," the overall style seems to have undergone minimal change. Can you provide an explanation for these observations?
>
> We have conducted an additional comparison with ProxNPI, illustrated in Figure 8(d) and thoroughly discussed in Section A.4 in the revised paper. This comparison reveals that NMG achieves results on par with those showcased in the ProxNPI study. However, it is essential to note that ProxNPI does not extensively explore various editing scenarios, such as contextual alterations or stylization. In contrast, NMG has demonstrated its proficiency across a wide range of editing scenarios.
>
> Additionally, as elaborated in Section 4.1, ProxNPI faces limitations due to its dependence on inversion guidance. NMG, free from such constraints, exhibits greater versatility and effectiveness in various image editing tasks, underscoring its broader applicability and utility in the field of image editing.
>
> Note that ProxNPI was an unpublished arxiv work at the time of our submission, and our work was independently developed in parallel(i.e., concurrent work). While ProxNPI may be a noteworthy project, we may believe it'd be unfair to us to be discounted by ProxNPI with a single perspective. Our research offers unique insights and contributions that stand on their own merit and we respectfully ask an unbiased evaluation by the reviewer, considering this matter.

---

> ### Author Response · Authors · 2023-11-23
> **Looking forward to your response**
>
> Dear Reviewer yyX5
>
> We appreciate your detailed and insightful review. As the rebuttal phase nears its conclusion, we are contacting you to inquire if there are any further questions or concerns we can clarify. Understanding the significant demands on your time and the importance of thorough deliberation, I would like to kindly request your insights and comments on our response. Thank you for your time and contribution to the peer review process.

---

> > ### Comment · Reviewer_yyX5 · 2023-12-04
> > **Response from Reviewer Reviewer yyX5**
> >
> > The rebuttal well solves my questions. I would recommend the acceptance of the paper.

---

### Author Response · Authors · 2023-11-21
**Official Comment by Authors**

Thank you for the reviewers' valuable insights and constructive feedback on our manuscript. We have thoughtfully considered suggestions and have made several significant improvements to enhance the clarity and depth of our work. Newly added sections are highlighted in blue for ease of reference.

**Improvement of Figure 2 (```reviewer 9KmB```)** We have revised Figure 2 to provide a more precise and comprehensive visual representation. This improved figure now better demonstrates the capabilities and advantages of our NMG, particularly in comparison to NTI. The enhancements in this figure aim to address the previously noted ambiguities and provide a more intuitive understanding of our method’s functionality.

**Additional Results of Section A.4 and Figure 8 in the Appendix(```reviewer yyX5``` ```reviewer 9KmB```  ```reviewer pVhT``` )** In response to the need for additional experimental results, we have included Figure 8 and an accompanying description in Section A.4. This section provides a thorough comparative analysis with ProxNPI, an in-depth analysis of combining NMG with NTI, and additional reconstruction results and user study findings.

**Inclusion of Section B in the Appendix (```reviewer yyX5``` ```reviewer 9KmB```  ```reviewer pVhT```)** Section B has been included to discuss our work's limitations and potential future directions. Here, we address challenges in editing tasks that require significant modifications to spatial structures. This section aims to provide a transparent and honest assessment of our method's capabilities and outlines areas for future research and development.

These updates have been made to address the key concerns and suggestions raised by the reviewers. We believe that these improvements significantly augment the quality of our manuscript, providing more precise insights into our method's strengths, limitations, and potential for future advancement.

---

### Public Comment · ~Senmao_Li2 · 2023-12-05
**Question about Sec.3.2 Noise Map Guidance**

This paper is highly intriguing and is poised to have a positive impact on the community. The paper is well-supported by evidence, with experiments that provide strong support for the claims presented. I have carefully read the paper and encountered some problems that I cannot quite understand:

1. In Sec. 3.2, the term $\triangledown_{z_t}\text{log}p(z_t)$ in Equation (7) seems to be following [1]. However, the paper does not explain how $\triangledown_{z_t}\text{log}p(z_t)$ is obtained. Is it approximated using $\epsilon_\theta$ ?

2. The term $c_N$ in Equation (9) appears not to be utilized on the RHS. Does $c_N$ represent $z^*_{t-1}$ ?

3. Why is it permissible to make the approximation $z^{NM}\_t$$\approx$$z^{NM}\_{t-1}$ in Equation (12) ? Does this approximation introduce errors? If errors are introduced, could it lead to inaccuracies in the reconstruction of real images?

Looking forward to receiving a response from the author, thanks very much.

**References**

[1] Scorebased generative modeling through stochastic differential equations. https://arxiv.org/pdf/2011.13456.pdf

---

### Meta-Review · Area_Chair_gJ9H · 2023-12-10

**Metareview:**

**Summary**

This paper proposes noise map guidance (NMG) for editing real images with text-guided diffusion models. The proposed method is optimization-free unlike the null-text inversion, and better preserves the spatial context of the input image during editing compared with existing real image editing methods.

**Strengthes**

The proposal looks simple yet effective. In particular it is optimization-free and spatial-context-aware. Extensive numerical experiments were conducted to demonstrate its effectiveness.
All the three reviewers evaluated this paper highly, above the acceptance threshold.

**Weaknesses**

The reviewers raised several concerns, which however seem to have been addressed appropriately in the revision.

Upon my own reading of this paper, I noticed the following minor points, which would require appropriate revision.
- Page 3, line 24: The expression "a reconstruction and editing sequence" would give an incorrect impression that there is only a single sequence. In fact there are two, a reconstruction sequence and an editing sequence, so that an appropriate wording is necessary.
- Page 5, line 16: due to (the observation? that) it gives us ...
- Page 5, lines 26, 27: Eq 9 $\to$ Eq. 9

**Justification For Why Not Higher Score:**

The proposal in this paper can be thought of as an instance of technical improvement of inversion with diffusion models for real image editing, so that the scope would not be wide enough to attract attention of larger audiences.

**Justification For Why Not Lower Score:**

This paper proposes a simple yet effective method for real image editing with conditional diffusion models. All the reviewers evaluated this paper above the acceptance threshold. This paper would also stimulate further studies on improving performance of real image editing.

---

### Decision · Program_Chairs · 2024-01-16

Accept (poster)